# Joint Encoding of KV-Cache Blocks for Scalable LLM Serving

## Abstract

Modern large language models (LLMs) drive interactive AI systems but are bottlenecked by the memory-heavy growth of key-value (KV) caches, which limits real-time throughput under concurrent loads. Existing KV-cache compression methods rely on rigid heuristics, disrupt tensor layouts, or require specialized compute, hindering scalability and deployment.

We propose joint encoding of KV-cache blocks, which fuses similar blocks across requests and input chunks into shared representations while preserving standard cache structure. This alleviates the KV-cache memory bottleneck, supporting high-concurrency serving without specialized hardware. Theoretically, we analyze the rate-distortion tradeoff of fused cache blocks under a Poisson process model. Empirically, our method achieves up to $4.38\times$ KV-cache compression with negligible accuracy loss across diverse LLMs and benchmarks, outperforming recent structured and adaptive compression baselines. In real LLM serving, joint encoding improves the token throughput by $\sim$40% on a single-machine vLLM benchmark, demonstrating substantial gains in inference throughput. Code is available at `kv_joint_encoding-55B0`.

## 1 Introduction

Large Language Model (LLM) services have recently gained tremendous popularity, but their serving pipeline faces distinct challenges across two stages. In the *prefill stage*, the model processes the input sequence. This stage is compute-bound but executed only once per request. In contrast, the *decode stage* generates the subsequent output tokens, is repeated for every token, and is primarily constrained by memory bandwidth. The bottleneck arises because each decoding step requires fetching large model parameters from memory to maintain fast computation (Xie et al., 2025). This implies that batch size can be increased to serve more concurrent users without increasing generation latency, up to the point where computation time aligns with memory access time and becomes the new bottleneck.

Scaling LLM services for high concurrency rapidly encounters memory constraints. On-chip memory is consumed by three main components: the *model weights*, the *inference activations*, and the *key–value (KV) cache*. The weights, though large, are static and do not grow with users concurrency. Activations are lightweight, scale linearly with the number of users, and vanish after each step. The KV-cache, however, is both persistent and expansive: it scales with the number of active sessions, and requires reserving memory proportional to the maximum sequence length per user. In practice, the KV-cache rapidly becomes the dominant memory consumer, and hence, it is the primary obstacle to scaling users concurrency.

To alleviate these memory constraints, recent serving frameworks borrow ideas from virtual memory and paging in operating systems. In particular, vLLM introduces *Paged-Attention (PA)* Kwon et al. (2023a), which partitions the KV-cache of each request into fixed-size blocks, each holding a predefined number of key-value tokens. These blocks do not need to be contiguous in memory; instead, a *block table* tracks their locations, allowing the attention mechanism to retrieve relevant entries efficiently (Kwon et al., 2023b). This design removes the need of reserving per-user memory proportional to the maximum sequence length. Instead, memory consumption scales with the actual demand, which is typically far smaller on average.

Although Paged-Attention mitigates memory fragmentation, it does not address the fundamental asymmetry between the compute-bound prefill stage and the memory-bound decode stage. To address this, the vLLM scheduler prioritizes *prefill requests*, reducing the Time-To-First-Token (TTFT) and allowing larger decode batches. Yet, this prioritization comes at a cost of delaying *decode requests*, increasing the Time-Between-Tokens (TBT). In long-context scenarios, this elevated TBT can become severe, stalling generation for several seconds (Agrawal et al., 2023). Recently, studies suggested to separate the two stages: requests are first processed on prefill servers, and their KV-cache is then migrated to decode servers. This separation enables independent optimization of the stages, reducing TTFT during prefill and TBT during decoding (Qin et al., 2024; Zhong et al., 2024; Hu et al., 2024).

In the prefill phase, strategies such as *Tensor Parallelism* and *chunked-prefilling* have been shown to reduce TTFT and increase compute utilization (Agrawal et al., 2023; 2024). In the decode phase, however, compute utilization remains low because each step requires fetching the model parameters to generate only a single token per request. Increasing the batch size can mitigate this inefficiency, but larger batches demand proportionally more KV-cache and are limited by the memory.

To reduce this overhead, prefix sharing has been proposed: grouping requests with identical prefixes so they can share the cached blocks and hence minimize memory (Zhu et al., 2024; Juravsky et al., 2024). Yet, exact prefix matches are rare in heterogeneous workloads, limiting its practical benefit. This limitation motivates the need for more general block-sharing mechanisms that extend the advantages of prefix sharing without relying on exact prefix matches.

In this paper, we address the KV-cache bottleneck in LLM serving by introducing a *joint-encoding scheme* that compresses and reuses similar cache blocks across requests. We develop two complementary methods: *Batch Fast-Fusion (BFF)*, which fuses blocks *across different requests* prior to decoding, and *Chunks Fast-Fusion (CFF)*, which fuses blocks *across input chunks* during prefilling. Both schemes reduce KV-cache size, enable larger batch sizes, and lower network bandwidth demands. At the core of our approach is a *tree-structured fusion strategy* that efficiently identifies encoding opportunities while preserving model accuracy. By applying joint encoding in both prefill and decode phases, our scheme delivers high throughput even under heterogeneous workloads with diverse input/output lengths and arrival patterns. These new fusion strategies substantially extend the generality and efficiency of cache sharing, delivering both memory and bandwidth savings in heterogeneous, real-world workloads.

The remainder of this paper is organized as follows: Section 2 discusses the most relevant related work. Section 3 outlines the motivation behind block fusion in LLM serving systems. Section 4 describes the proposed scheme in detail. Section 5 analyzes the scheme in lens of the point processes, Section 6 presents the experimental results and evaluation, and Section 7 concludes the paper.

## 2 Related Work

A major line of research has focused on *reducing the memory footprint of the KV-cache*, primarily through compression. Key approaches include quantization, low-rank approximation, and selective eviction. *Quantization-based methods*, such as Hooper et al. (2024); Liu et al. (2024d), compress key and value embeddings to 2–3 bits with minimal accuracy loss, substantially increasing batch size and throughput over standard precision. Low-rank and latent representations, for example, Multi-Head Latent Attention (MLA) Liu et al. (2024a) and ReCalKV Yan et al. (2025) stored KV matrices in compact subspaces that are later reconstructed as needed. This approach achieves a significant reduction in the KV-cache size while maintaining attention accuracy. Recently, Meng et al. (2025) suggested a method for transforming the attention of pretrained models to MLA architectures.

Given the redundancy of KV states *across adjacent layers*, other works interpolate vector directions or use SVD to compress cross-layer states (e.g., Liu et al. (2024b); Chang et al. (2025)), often retaining only a subset of tokens for quality. Combining quantization with these techniques achieves KV-cache compression ratios up to 5×. Recent work leverages adaptive arithmetic encoding (Liu et al., 2024c), head-level token importance with residual merging (Liu et al., 2025), and sensitivity-driven dynamic sparsity (Zhang et al., 2024; Yang et al., 2025) to further boost cache efficiency.

A complementary research direction explores *memory sharing across requests* by reusing computation for common input segments. Frameworks like HydraGen (Juravsky et al., 2024), RelayAttention (Zhu et al., 2024), and vLLM/SGLang (Wu et al., 2025; Zheng et al., 2023) exploit prefix sharing, where requests with identical prefixes share cached states, reducing both computation and memory requirements. However, such methods are limited by the rarity of exact prefix matches in practical workloads. Fragmentation and fine-grained reuse of overlapping segments are considered in (Zhang et al., 2025; Prabhu et al., 2025; Yang et al., 2025).

Despite significant progress, most prior methods rely on rigid heuristics, disrupt native tensor layouts, or require exact prefix matches, limiting scalability and flexible integration into modern LLM serving pipelines. Our work directly addresses these limitations by introducing joint compression of KV-cache blocks, allowing finer-grained, layout-preserving, and general block sharing that extends the benefits of prefix reuse to arbitrarily similar segments. This flexibility is critical for memory- and bandwidth-efficient serving in real-world settings.

## 3 Motivation

Improving the throughput of LLM serving is often achieved by increasing batch size. Since the decoding phase is primarily memory-bound, dominated by repeated fetching of large weight matrices, larger batch sizes do not substantially increase per-request decoding latency. However, serving scalability remains constrained by the memory footprint of the KV-cache, which must be reserved per user session. Apart from the common compression methods like quantization and sparsification (by eviction), prefix-sharing techniques like Juravsky et al. (2024) and Zheng et al. (2024) demonstrate that overlapping prompt segments across requests enable KV-cache sharing. In addition, reuse of computed blocks as well as improved computation can improve throughput by up to 36% (Huijong Jeong & Kim, 2024). Nonetheless, these approaches rely on *exact* prefix matches, which severely limits applicability in real-world settings where variations in phrasing or task-specific inputs are common (Wu et al., 2025; Zheng et al., 2024). For example, two translation requests with slightly different introductory phrases ("Help me translate" vs. "Translate this") would fail to share any KV blocks under strict prefix matching (Wu et al., 2025).

Empirically, in Figure 1(a), we observe that KV-cache blocks occupy a large fraction of memory while exhibiting substantial redundancy. For Llama3.1 8B on nVidia OpenMathInstruct-2 (2024), more than 94% of sampled blocks have cosine similarity above a high threshold of 0.8, indicating many near-duplicate representations. At the same time, exact-prefix sharing opportunities are rare in heterogeneous workloads: only a small fraction of requests share an identical prefix, whereas a much larger fraction of blocks admits a close match elsewhere in the batch. This gap suggests that a similarity-based sharing mechanism can unlock significantly more reuse than prefix sharing alone, without changing the underlying attention computation.

To overcome the strict requirement of an exact match, we propose a KV-cache blocks joint-encoding scheme that fuses blocks based on similarity threshold. This approach extends the benefits of prefix sharing to non-identical contexts. Figure 1 highlights the memory and computational benefits of jointly encoding blocks in Softmax Attention (SA). Specifically, Figure 1(b) illustrates the benefits in the decoding phase, where keys with similar representations (indicated by color) can be combined into a unified representation. Fusing these keys (and values) into a unified representation not only allows increasing the batch size but also facilitates optimized matrix-matrix multiplication, instead of multiple matrix-vector multiplications.

In the prefill phase, Figure 1(c) illustrates how chunking input requests allows joint encoding of blocks across chunks, which reduces KV blocks memory, and thus, facilitates handling longer inputs and further reusing computation in SA. Consequently, any SA computation involving a fused block can be reused in subsequent chunks that include instances of this block, maximizing efficiency and throughput.

In both cases, after encoding, some blocks share the same representation and the block table points to the shared representation. Thus, any SA computation that involves a fused block can be reused in later chunks that comprise an instance of this block. The final SA result is derived by merging the SA of the fused components with that of the unique components (and rescaling). The recent version of vLLM can take advantage of the computation of SA with shared blocks.

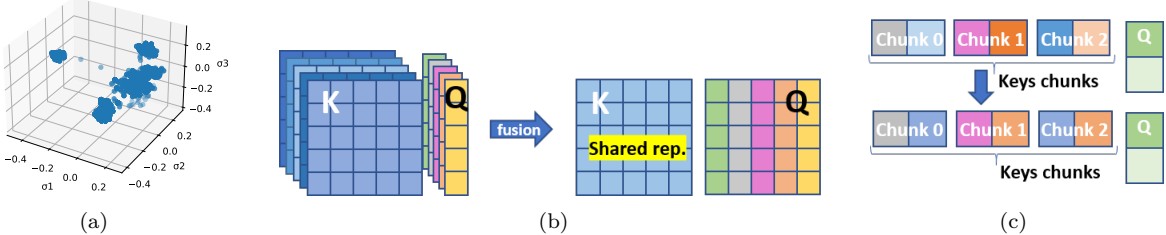

Figure 1: (a) The top three principal components of $\sim$ 7,000 blocks, each with 128 tokens from the nVidia OpenMathInstruct-2 (2024) dataset, in Llama3.1 8B. The cluttered patterns indicate substantial redundancy. (b) The effect on attention for a batch of requests with similar keys blocks. After fusion, the KV-cache footprint is reduced and batch decoding is optimized by performing matrix-matrix multiplication. (c) After fusion, the blocks computation of chunks 0 and 1 can be reused in chunk 2 (same color).

## 4   KV Blocks Joint-Encoding

In this section, we introduce our KV blocks joint-encoding scheme for the chunk-prefilling setting, to further facilitate the joint encoding of the chunks while they are processed in the LLM (Qin et al., 2024; Agrawal et al., 2023). The main objective of our scheme is to maximize the throughput by jointly encoding KV blocks. The impact is threefold. First, in the decode phase, benefit from a higher arithmetic efficiency due to a larger batch size, and further, minimize KV-cache prefetching, mitigating the memory bandwidth limit (see Figure 1(b)). Second, it also enables the reuse of fused blocks computations, alleviating the compute limitation (see Figure 1(c)). Third, reduce the usage of network bandwidth, which often becomes a bottleneck, especially when considering a high degree of parallelism (Agrawal et al., 2024). Certainly, it is crucial to achieve this goal while maintaining the model's accuracy.

To fulfill this goal, we propose Fast-Fusion (FF) method that jointly encodes similar blocks into a unified representation if the cosine similarity between contexts of different blocks is above predefined threshold. The similarity threshold is substantial for both the resulting *compression ratio* and *model's accuracy*. To avoid excessive blocks comparisons overhead, we design an efficient *tree-like fusion strategy* that scales as $O(B \log B)$ and $O(C \log C)$, where $B$ is the batch size and $C$ is the number of chunks, respectively. This strategy allows the parallelization of the fusion process at each level of the tree. Accordingly, the fusion can be done over blocks of different requests (i.e., BFF) or blocks of different chunks (i.e., CFF).

In practice, the KV-cache layout in vLLM is $(B, p, t, h, d)$, where $p, t, h$, and $d$ are the number of blocks, tokens per block, number of heads, and the head embedding size, respectively. Before encoding, the KV-cache is unfolded into a convenient layout of blocks per request or chunk. Specifically, let $r = t \cdot h \cdot d$, and note that $C = \lfloor p \cdot t/(\text{size of chunk}) \rfloor$, we use a layout of $(B, p, r)$ and $(C, p/C, r)$ for the BFF and the CFF, respectively. The norm of each $r$ is stored to allow a proper rescaling of the fused blocks. The algorithm is applied iteratively for every $N$ chunks or requests in each layer.

A pseudo-code of the FF algorithm is given in Algorithm 1. Roughly speaking, after unfolding the KV-cache into a convenient layout and storing the norms, we recursively call the FF method in Algorithm 1, which fuses blocks of different requests or chunks if their similarity level is above threshold. Intuitively, blocks represented by $r$-dimensional vectors can also be expressed in terms of their norm and corresponding $r$-dimensional unit (direction) vectors. In this view, fusion can be understood as aligning multiple unit vectors into a single unified direction, while *preserving the distinctiveness of the original blocks through their norms*. This allows representing multiple blocks using a single unit vector and a norm (scalar) per block. To further enhance the compute, the fused blocks are taken into account in the attention computation, allowing low-level kernels to leverage jointly encoded blocks in the decoding phase, and reuse computations in the prefill phase. Only one copy of the fused blocks is needed, and redundant copies can be evicted to reduce memory usage. Further, different number of blocks may be encoded in each layer, which requires running each layer with its own

block table, as already implemented in vLLM. Note that our scheme attains a compression that is at least the compression attained by shared prefix methods, since we further compress inputs that do not share prefixes.

Consequently, the joint encoding allows us to reuse computations to address the limitations of compute, memory bandwidth, and network bandwidth. The tree structure is beneficial for detecting fusion opportunities at a reasonable cost in large-scale prefill and decoding serving systems.

*Remark* 4.1. The order of requests impact the resulting compression. Specifically, it is beneficial to place shorter requests (in terms of number of blocks) on the left tree to gain higher diversity per block. At the best case, all the blocks in the right tree will be unified into the left tree blocks representation, yielding the maximal compression. However, the responsibility of ordering the requests belongs to the scheduler, which is out of the scope of this paper.

*Remark* 4.2. To avoid manual tuning, the threshold can be self-adjusted using a simple feedback. For example, percentile-based or target-compression strategies monitor fusion statistics during serving: if too many blocks are fused (or compression exceeds a desired CR), the threshold is increased, and vice versa. Alternatively, a small calibration set can be used to refine the threshold based on acceptable distortion levels, providing a lightweight and fully automated adaptation mechanism.

*Remark* 4.3. Prior work (e.g., Wu et al. (2025)) demonstrated that prefix-sharing may leak information through timing side-channels. In particular, by examining the TTFT, an attacker can determine whether a particular prefix has already been cached by the vLLM. Interestingly, joint encoding mitigates such risks: because *multiple distinct KV blocks* are mapped to a shared latent direction, any reduction in compute time is no longer attributable to a single prefix or token, but to a many-to-one cluster. This substantially weakens the leakage signal compared to prefix matching.

---

**Algorithm 1** Fast Fusion

**Input:** KV-cache (k,v), threshold (*thr*)
**if** len(k) == 1 **then**
    **return** normalize(k), normalize(v)
**end if**
(*tree left*)
$k_l, v_l \leftarrow$ FastFusion(k[: len(k)//2], v[: len(v)//2], *thr*)
(*tree right*)
$k_r, v_r \leftarrow$ FastFusion(k[len(k)//2 :], v[len(v)//2 :], *thr*)
$\text{sim} \leftarrow (k_l k_r^{\top})$
(*similarity per block of $k_l$*)
**for** each *row* in sim **do**
    $idx_{\text{fused}} \leftarrow$ indices where $\text{sim}(row) > thr$
    **if** $|idx_{\text{fused}}| > 0$ **then**
        (*block fusion*)
        $k_l(row) \leftarrow$ normalize($k_l(row) + \sum_{i \in idx_{\text{fused}}} k_r(i)$)
        $v_l(row) \leftarrow$ normalize($v_l(row) + \sum_{i \in idx_{\text{fused}}} v_r(i)$)
    **end if**
**end for**
update ref. count of blocks and update block-table
**return** cat([$k_l$ , $k_r$ unique blocks]), cat([$v_l$ , $v_r$ unique blocks])

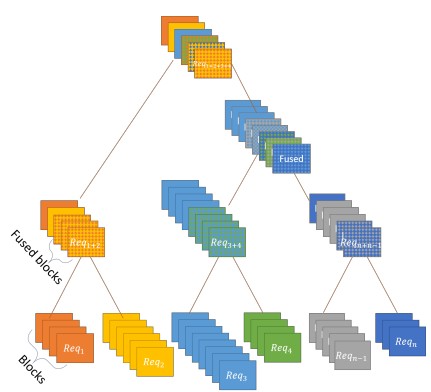

Figure 2: BFF example for 6 request, each has a different number of blocks (and a different color). The Fast Fusion is recursively called, fusing every pair of requests (depicted by a mixed color). Then, fusing pairs of pairs, and so on, until all the requests' blocks are jointly described in the KV-cache. Clearly, some blocks are not fused (original color), and some are fused more than once while traversing the tree.

## 5 Analysis

The similarity threshold is substantial to the algorithm performance as it dictates the resulting accuracy (or distortion) on the one hand, and the compression (or rate) on the other. In terms of rate-distortion, high threshold ensures low KV cache distortion, yet may result in low exceedance rate, and consequently, moderate compression. Similarly, low threshold value allows unifying many blocks into a single representation (high compression), yet may degrade the performance due to a highly distorted KV cache. Hence, it is crucial to thoroughly study the impact of the threshold.

To analyze the resulting rate, recall that Algorithm 1 recursively fuses pairs of requests (see Figure 2 for illustration). Assume a pair of requests with $m_1$ and $m_2$ blocks, respectively, for which there are $n = m_1 \cdot m_2$ similarity samples $\{x_i\}_{i=1}^n$. Then, using Kernel Density Estimation (KDE) with a Gaussian kernel (Wand & Jones, 1994, Ch. 2), the similarity density in each layer is approximately $f_h(x) = \frac{1}{nh} \sum_{i=1}^n \phi\left(\frac{x-x_i}{h}\right)$, where $h$ is the kernel bandwidth (standard deviation). Therefore, the similarity distribution is approximately

$$F_h(x) = \frac{1}{n} \sum_{i=1}^n \Phi\left(\frac{x - x_i}{h}\right), \tag{1}$$

where $\phi$ and $\Phi$ are the Gaussian density and distribution functions, respectively. In other words, each sample $x_i$ contributes a Gaussian kernel function with mean value $x_i$ to the overall estimated probability density function, thus contributing a different probability of exceeding the threshold.

**Proposition 5.1.** *For a sufficiently high similarity threshold $u$, the number of above-threshold similarities is asymptotically a Poisson variable with rate*

$$\Lambda(u) = \frac{1}{n} \sum_{i=1}^n \exp\left\{-\frac{u - (hb_n + x_i)}{ha_n}\right\}, \tag{2}$$

*where $a_n = (2\log n)^{-1/2}$ and $b_n = (2\log n)^{1/2} - \frac{1}{2}(2\log n)^{-1/2}(\log\log n + \log(4\pi))$ are normalization constants.*

*Proof.* Our goal is to analyze the asymptotic threshold exceedance rate of eq. (1). Since the indices of high threshold exceedance are random and are relatively rare, the number of above-threshold observations can be modeled as a Poisson random variable when the threshold is sufficiently high (Leadbetter et al., 2012, Ch. 5). In particular, let $u$ be a threshold such that the kernel of each sample $i$ satisfies $\Pr(\mathbf{x}_i > u) = \left(1 - \Phi\left(\frac{u-x_i}{h}\right)\right) = \Theta(1/n)$. Then, according to the *uniformly asymptotically negligible* condition, the number of threshold arrivals is approximately a Poisson variable with rate $\Lambda(u)$ when the samples are independent (Falk et al., 2010, Ch. 8.3). Similar treatment for dependent samples is given in (Coles et al., 2001, Ch. 5).

For the Gaussian case, the threshold exceedance rate is $\Lambda(u)$ given in eq. (2), where $a_n = (2\log n)^{-1/2}$ and $b_n = (2\log n)^{1/2} - \frac{1}{2}(2\log n)^{-1/2}(\log\log n + \log(4\pi))$ are normalizing constants (Kampeas et al., 2014, Theorem 5). □

Figure 3(a) depicts the empirical similarity and the above-threshold distribution together with the KDE approximation for DeepSeek-R1-Distill-Qwen-7B (2025) on nVidia HelpSteer (2024) dataset in several layers. Evidently, for this model and dataset, the similarity appears Gaussian, yet with different mean and variance in each layer. Interestingly, letting the similarity in layer $\ell$ be approximately Gaussian with mean $\mu_\ell = \frac{1}{n}\sum_{i=1}^n x_i$ and variance $\sigma_\ell^2 = \frac{1}{n}\sum_{i=1}^n x_i^2 - \mu_\ell^2$, by the Poisson point process, the threshold exceedance rate is approximately $\Lambda_\ell(u) = n \cdot \left(1 - \Phi\left(\frac{u-\mu_\ell}{\sigma_\ell}\right)\right)$. Moreover, even though each layer follows a different distribution and, therefore, has a different threshold exceedance rate, by Proposition 5.1, the overall exceedance rate for all model layers can be evaluated.

The Poisson formulation facilitates analyzing the probability of observing non-compressible layer, and the expected compression ratio over the entire model. From the Poisson distribution properties, we have the following corollary.

**Corollary 5.2.** *After the fusion, the compression ratio over $L$ layers is*

$$\text{compression ratio} = L(m_1 + m_2) / \left((L(m_1 + m_2) - \sum_{\ell=1}^L \Lambda_\ell(u)\right).$$

*The probability of no fusion in layer $\ell$ is*

$$\Pr(\text{no fusion in layer } \ell) = \exp(-\Lambda_\ell(u))$$

To analyze the resulting distortion, let us examine the attention distribution drift that stems from fusing blocks. Let $\mathbf{z}_t = \mathbf{q} \cdot \mathbf{k}_t / \sqrt{d}, \forall t \in 1, \dots, p$ be the logits vector of a block with $p$ tokens, and let let $\mathbf{s} = \exp(\mathbf{z}) / \sum_j \exp(z_j)$

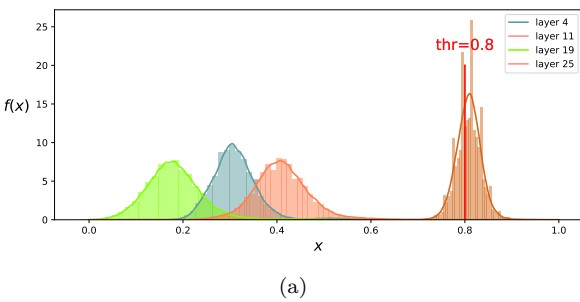 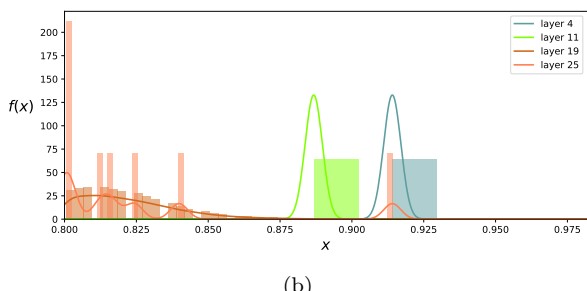

(a)                                           (b)

Figure 3: Empirical similarity and analysis for DeepSeek-R1-Distill-Qwen-7B (2025) on nVidia HelpSteer (2024) dataset for several layers. Bars represents the empirical similarity and solid line the KDE approximation. (a) Blocks similarity density. (b) Above-threshold density for threshold=0.8. This empirical exceedance rate aligns with the Poisson approximation for high thresholds.

be the softmax distribution before fusion. Similarly, let $\mathbf{z}' = \mathbf{z} + \delta$ and $\mathbf{s}' = \exp(\mathbf{z}')/\sum_j \exp(z'_j)$ be the perturbed (fused) logits and softmax distribution. Then, the attention distribution drift satisfies the following bound.

**Proposition 5.3.** *After fusion, the attention distribution drift is bounded by*

$$\|\mathbf{s}' - \mathbf{s}\|_1 \leq 2\epsilon + O(\epsilon^2)$$

*where $\epsilon = \max_i \frac{\|\mathbf{k}_i\|\|\mathbf{q}\|}{\sqrt{d}}\sqrt{2(1-u)}$.*

*Proof.* First, since $\mathbf{k}$ and $\mathbf{k}'$ have the same norm, we can represent then as $\mathbf{k} = \|\mathbf{k}\| \cdot \hat{\mathbf{k}}$ and $\mathbf{k}' = \|\mathbf{k}\| \cdot \hat{\mathbf{k}}'$. Let $\delta_t = \mathbf{z}'_t - \mathbf{z}_t = \|\mathbf{k}\| \cdot \mathbf{q} \cdot (\hat{\mathbf{k}}' - \hat{\mathbf{k}})/\sqrt{d}$. Thus,

$$\begin{aligned}
\|\delta_t\| = &\|\mathbf{z}'_t - \mathbf{z}_t\| \\
\leq &\|\mathbf{k}\| \cdot \|\mathbf{q}\| \cdot \|\hat{\mathbf{k}}' - \hat{\mathbf{k}}\|/\sqrt{d} \\
= &\|\mathbf{k}\| \cdot \|\mathbf{q}\| \cdot \sqrt{2(1 - \hat{\mathbf{k}}' \cdot \hat{\mathbf{k}})}/\sqrt{d} \\
\leq &\|\mathbf{k}\| \cdot \|\mathbf{q}\| \cdot \sqrt{2(1 - u)}/\sqrt{d}
\end{aligned}$$

Since $\epsilon = \max_i |\delta_i|$, the denominator of $\mathbf{s}$ satisfies

$$\exp(-\epsilon)\sum_j \exp(z_j) \leq \sum_j \exp(z_j + \delta_j) \leq \exp(\epsilon)\sum_j \exp(z_j)$$

Similarly, for the numerator of $\mathbf{s}$ satisfies

$$\exp(-\epsilon)\exp(\mathbf{z}) \leq \exp(\mathbf{z} + \delta) \leq \exp(\epsilon)\exp(\mathbf{z})$$

From these bounds we obtain that each index $i$ satisfies $\exp(-2\epsilon)s_i \leq s'_i \leq \exp(2\epsilon)s$, as

$$\mathbf{s}' \leq \exp(\epsilon)\exp(\mathbf{z})/\exp(-\epsilon)\sum_j \exp(z_j) = \exp(2\epsilon)\mathbf{s}.$$

Now, let us divide into the case. When $s'_i \geq s_i$, then $s'_i - s_i \leq s_i(\exp(2\epsilon) - 1)$. Similarly, when $s'_i < s_i$, then $s_i - s'_i \leq s_i(1 - \exp(-2\epsilon)) \leq s_i(\exp(2\epsilon) - 1)$. Thus, $|s'_i - s_i| \leq (\exp(2\epsilon) - 1)s_i$.

Finally, sum up the index $i$, since $\sum_i s_i = 1$, we have $\sum_i |s'_i - s_i| = \|\mathbf{s}' - \mathbf{s}\|_1 \leq (\exp(2\epsilon) - 1) \leq 2\epsilon + O(\epsilon^2)$ from Taylor expansion. Since $\epsilon \leq \max_i \|\mathbf{k}_i\|\|\mathbf{q}\|\sqrt{2(1 - u)}/\sqrt{d}$, the lemma follows. □

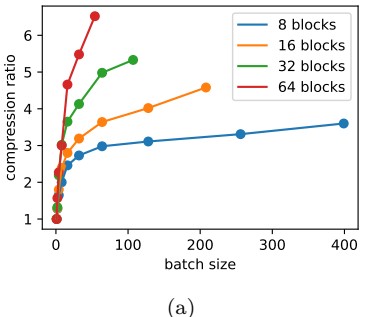 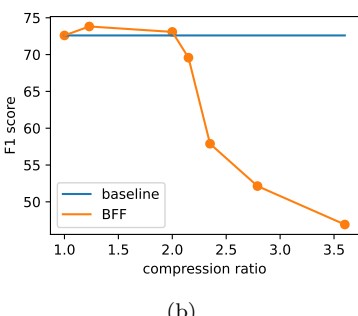 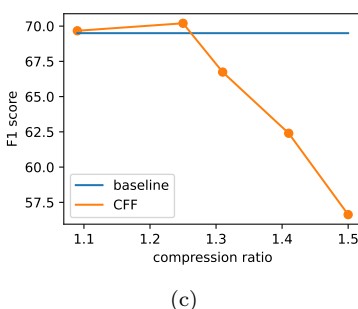

(a)            (b)            (c)

Figure 4: CR and F1 score of BFF and CFF for Llama-2 7B. (a) CR vs. batch size for diverse number of blocks on vLLM random-data benchmark. (b) BFF F1 score vs. CR for batch size 256 on conversational dataset. (c) CFF F1 score vs. CR, for 4 chunks on conversational dataset.

The analysis provides a theoretical foundation for understanding the trade-offs inherent in block fusion for LLM serving. The Poisson point process modeling indeed shows that the similarity threshold directly governs the balance between compression and distortion in the KV cache. Our per-layer analysis predicts the expected compression for a given threshold, allowing skipping non-compressible layers in probability.

# 6 Results

In this section, we present the results of our fusion scheme for CFF and BFF. Measuring the accuracy, compression, and latency together requires a dedicated kernel, which we keep in mind for future work. Sections 6.1-6.3 assess compression and accuracy, by fuse and replicate the rescaled cache across all fused blocks (without freeing blocks). Section 6.4 examines the end-to-end latency by fusing and freeing blocks (without rescaling).

We evaluate the performance of our approach using a commodity GPU on various benchmarks and models and compare it to the baseline performance. The results demonstrate the effectiveness of our scheduling algorithm in terms of compute efficiency, memory reduction, and network bandwidth consumption.

## 6.1 Block Diversity and Rate-Distortion

Both the number of requests (or chunks) and the number of blocks in each request (or chunk) influence the resulting Compression Ratio (CR). The larger the batch, the more requests to fuse, and the more blocks per request, the greater diversity of blocks for fusion, which yields a better CR. To assess the significance of each, we examine the BFF CR on the random data benchmark in vLLM API (2023), using a fixed similarity threshold for a various number of requests and blocks per request using Llama2-7B with 16 tokens per block. Interestingly, Figure 4(a) reveals that the diversity of blocks per request is more significant than the diversity of requests, since the CR grows faster when more blocks per request are used.

Figure 4(b) and 4(c) depict the the rate distortion trade-off for BFF and CFF, respectively. The rate distortion curve is given for Llama2-7B on Das (2024) conversational dataset, where the number of requests (chunks) is fixed, and the similarity threshold varies. The baseline result is given for reference. Specifically, in Figure 4(b) the BFF achieves a CR of $\sim 2.15\times$, without degrading accuracy for batch size 256 with a block size of 16 tokens. In Figure 4(c), the CFF achieves a CR of $\sim 1.25\times$ without losing accuracy for 4 chunks with a block size of 16 tokens.

Remarkably, this rate distortion formulation indicates that a higher CR can be achieved without sacrificing accuracy when fixing the similarity threshold value. In particular, since only the threshold determines the resulting accuracy (distortion), once the threshold is fixed, increasing the batch size or the number of blocks per request yields a better CR due to diversity in requests and blocks.

## 6.2 Batch Fast-Fusion During Decoding

The impact of diversity on the CR and the resulting F1 score is of great practical interest, as it indicates the gain of a larger batch using our enhancement. Specifically, in this section, we investigate the CR and the F1 score when increasing the number of requests in the BFF scheme, where the similarity threshold is set to a fixed value, using the nVidia OpenMathInstruct-2 (2024) dataset.

Figure 5(a) depicts the CR versus batch size for the Llama3.1-8B model and Qwen2.5-72B. Notably, in both cases, the CR grows logarithmically with the batch size, reaching CR of $\sim 3.11\times$ and $\sim 4.38\times$ for batch sizes of 128 and 64, respectively.

Using the same settings, we further evaluate the behavior of the F1 score when increasing the number of requests in a batch in Figure 5(b). Interestingly, the F1 score is a bit higher on average for both models, and especially for the Qwen2.5 72B model. The phenomenon where averaging similar blocks improves the model accuracy can be interpreted through the lens of the *crowd wisdom effect*. Each block representation, akin to an individual expert, contributes unique insights about the block context over the layers. When focusing on relatively similar representations, averaging these representations reduces individual biases and errors, much as a group makes more accurate decisions (Trott, 2024).

Table 1 describes the F1 score and the CR behavior when applying BFF to every 8 requests (i.e., batch size 8) for various thresholds in a variety of MMLU (2020) tasks and GSM8k (2022). These results highlight the ability of the BFF to significantly reduce the KV cache size in many cases, without compromising accuracy, showcasing its value in improving decoding efficiency.

Table 1: F1 score and CR (in parenthesis) achieved by BFF for Llama3.1-8B.

| Model | Method | GSM8K | Con.Phy. | E.Eng. | F.Logic | HS.Bio | misc. | sociology | Average |
|---|---|---|---|---|---|---|---|---|---|
| | BFF | 41.47 | 37.87 | **40.69** | 35.2 | 44.19 | 51.72 | **40.8** | 41.71 |
| | thr=0.7 | (×3.1) | (×3.29) | (×3.21) | (×2.69) | (×2.51) | (×2.93) | (×3.11) | (×2.98) |
| | BFF | 51.91 | 40.85 | 38.62 | 40 | 46.13 | 51.34 | 39.3 | 44.02 |
| Llama3.1-8B | thr=0.74 | (×2.52) | (×2.52) | (×2.48) | (×2.21) | (×2.04) | (×2.32) | (×2.4) | (×2.36) |
| | BFF | **62.86** | 40.43 | 40 | **41.27** | 45.48 | **54.79** | 38.31 | 46.16 |
| | thr=0.78 | (×2.09) | (×1.63) | (×1.59) | (×1.53) | (×1.37) | (×1.52) | (×1.61) | (×1.62) |
| | Baseline | 62.46 | **40.85** | 40.69 | 40 | **48.39** | 54.79 | 40.8 | **46.85** |

'

Practically, these results indicate up to $\sim 3.11\times$ and $\sim 4.38\times$ reduction of KV cache blocks, for Llama3.1-8B and Qwen2.5-72B, respectively. Of course, this also mitigates the memory fetching and the network bottleneck which is significant, especially when considering a high level of parallelism. Furthermore, using fused blocks is beneficial for performing hardware-optimized matrix-matrix multiplications (Juravsky et al., 2024).

## 6.3 Chunks Fast Fusion During Prefill

In this section, we examine the CR and the resulting F1 score for the CFF. Even though eliminating the distance that stems from RoPE can yield a higher CR, it disables reusing computations, which is substantial for the prefill phase. Thus, the CFF is applied to blocks within chunks, together with their positioning. In addition to compression, the results also indicate the computation reuse factor.

To characterize the impart of CFF on the CR and F1 score, the similarity threshold is set to a fixed value, and the number of chunks to fuse is varied. The evaluation is performed on Llama3.1-8B and Qwen2.5-72B models with 16 tokens per block on the Longbench qmsum dataset Bai et al. (2023). This dataset contains relatively long inputs, which allows characterizing the CR and accuracy of CFF when applied to increasing number of chunks. Figure 6(a) depicts the CR and when applying CFF to the chunks. As we see, the CFF uses up to $\sim 3.25\times$ fewer blocks, for which their computation can be reused, thus mitigating computational and network bottlenecks. Figure 6(b) depicts the F1 score of the CFF when scaling the number of chunks to fuse on the same task. The CFF manages to keep the accuracy in most cases, and experiences only a negligible accuracy loss.

Table 2 describes the F1 score and the CR behavior when applying CFF to every 8 chunks for various thresholds in a variety of Longbench tasks (Bai et al., 2023). The table highlights consistent gains in

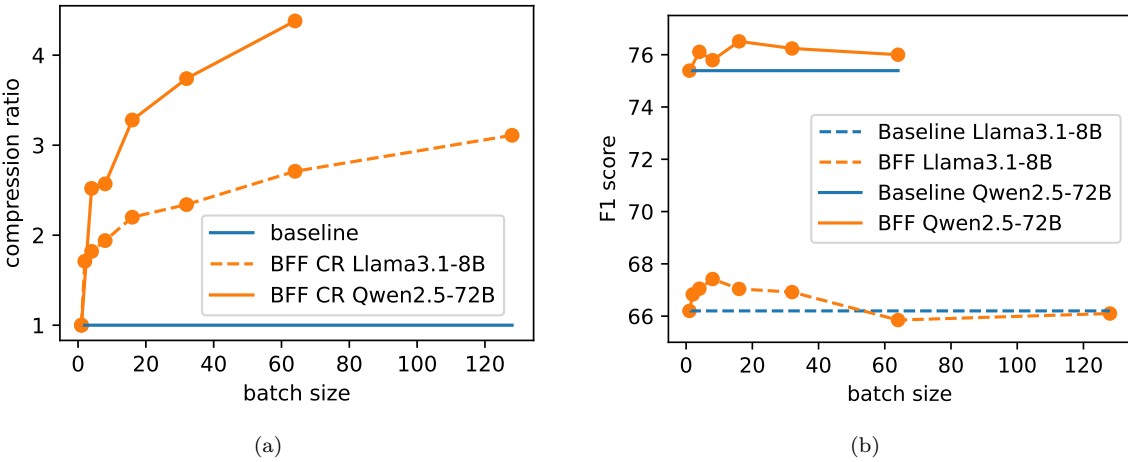

(a)            (b)

Figure 5: CR and F1 score of BFF vs. baseline for Llama-3.1 8B and Qwen2.5 72B on nVidia OpenMathInstruct-2 (2024) dataset. (a) CR vs. batch size. (b) F1 score vs. batch size.

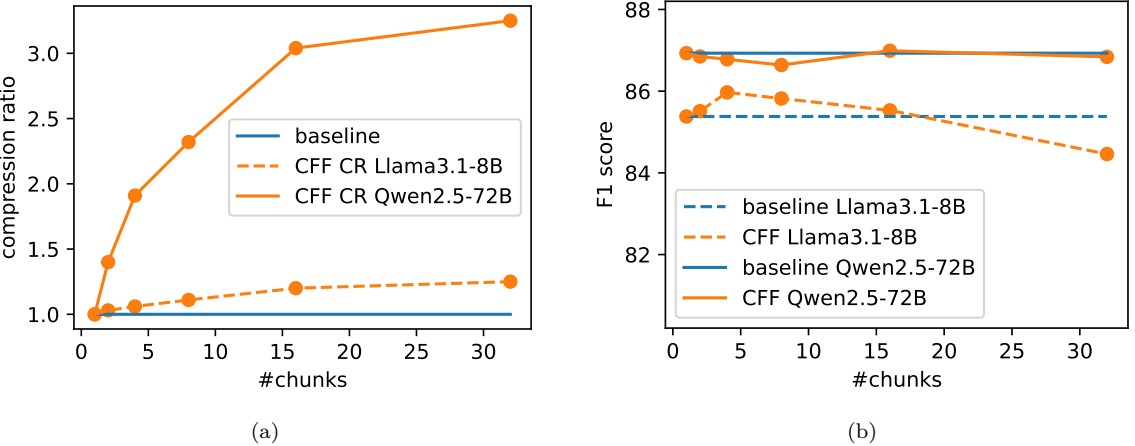

(a)            (b)

Figure 6: CR (a) and F1 score (b) of CFF vs. number of chunks for Llama3.1-8B and Qwen2.5-72B using the Longbench qmsum dataset.

compression with minimal or no loss in accuracy. Notably, even at lower thresholds, the F1 scores remain comparable to the baseline, while achieving compression ratios of up to ×1.87.

Table 2: F1 score and CR (in parenthesis) achieved by CFF for Llama3.1-8B.

| Model | Method | LCC | RepoBench-P | PR-en | TREC | 2wikimqa | GovReport | MQA-zh | Average |
|-------|--------|-----|-------------|-------|------|----------|-----------|--------|---------|
| | CFF | 75.77 | 74.07 | **21.04** | 40.48 | **45.9** | 83.8 | 27.34 | 52.63 |
| | thr=0.62 | (×1.87) | (×1.59) | (×1.32) | (×1.43) | (×1.38) | (×1.59) | (×1.53) | (×1.53) |
| | CFF | 77.12 | 74.8 | 18.3 | 40.3 | 45.88 | 81.39 | 38.08 | 53.7 |
| Llama3.1-8B | thr=0.64 | (×1.49) | (×1.3) | (×1.15) | (×1.16) | (×1.17) | (×1.25) | (×1.24) | (×1.25) |
| | CFF | 77.24 | **75.99** | 18.38 | **41.15** | 44.94 | **83.83** | **40.31** | **54.55** |
| | thr=0.66 | (×1.26) | (×1.15) | (×1.06) | (×1.06) | (×1.06) | (×1.09) | (×1.1) | (×1.11) |
| | Baseline | **77.73** | 75.57 | 18.74 | 40.95 | 44.12 | 82.81 | 39.18 | 54.16 |

'

### 6.4 End-to-End Serving Perfromance

We evaluate BFF on a single-machine vLLM serving benchmark over a random dataset of 100 requests with input size 1024 and varying number of output tokens (with random-ratio 0.5) using Llama-3.1-8B. After the fusing blocks (with similarity above threshold 0.7), the model runner sends the scheduler the updated block-tables, allowing the scheduler to free unused blocks, and increase reference counts for shared blocks for further reuse across requests.

Figure 7 depicts the throughput and end-o-end performance achieved by our naive implementation, where the entire fusion overhead is reflected in the decode phase. Remarkably, the BFF yields measurable system-level benefits: lower prefill latency (TTFT), higher effective batching, and increased throughput. The Decode-phase inter-token latency (ITL) increased due to larger batch GEMMs, and the fusion overhead. However, the overall serving throughput and request completion time reduced substantially.

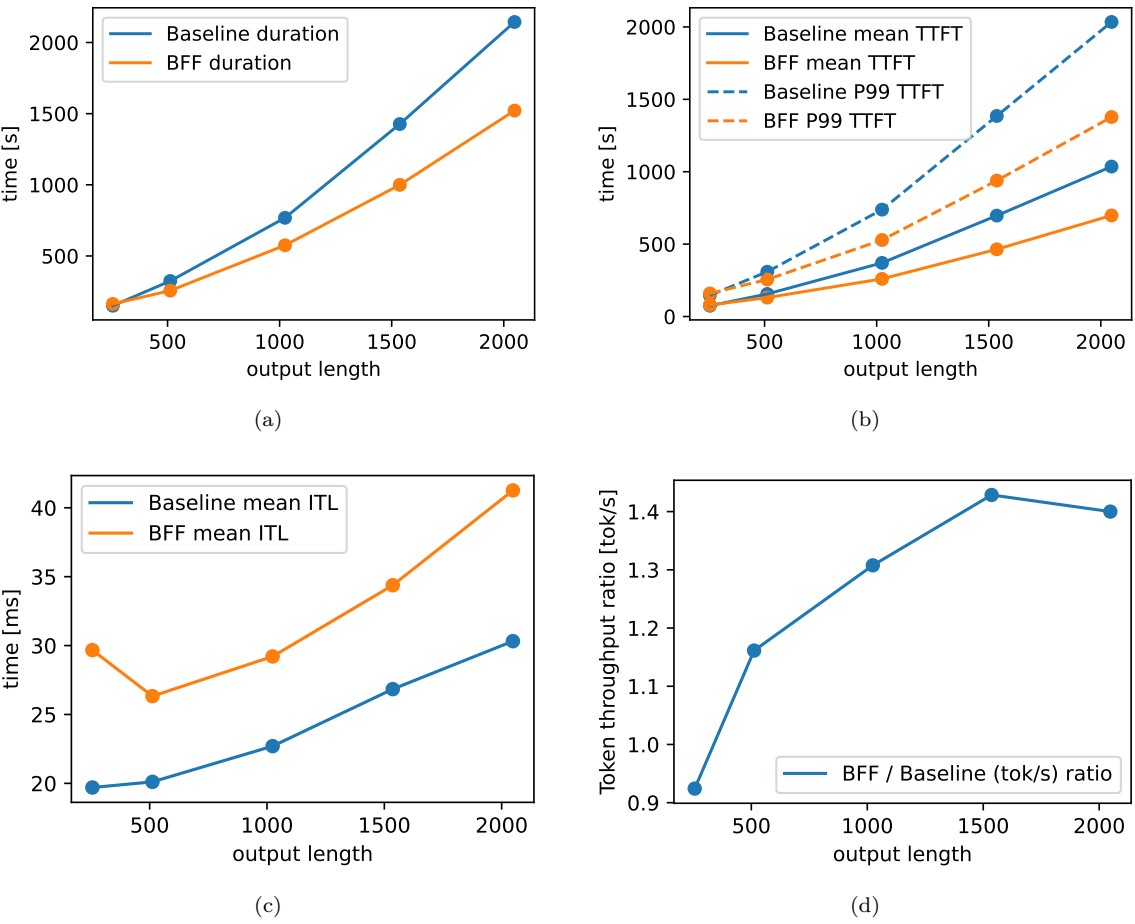

Figure 7: Throughput and end-to-end performance comparison on Llama3.1-8B (a) benchmark duration (b) TTFT mean and P99 (c) mean ITL (d) throughput ratio (tok/sec)

Overall, the results presented in this section demonstrate the effectiveness of our context-sharing scheduling scheme for the KV cache-centric disaggregated architecture. Our approach significantly improves compute efficiency, reduces memory and network bandwidth consumption, and scales well with increasing system size, making it a promising solution to accelerate LLM serving in resource-constrained scenarios.

# 7    Conclusion

In this paper, we presented *Fast Fusion*, a novel context-sharing enhancement that can improve LLM serving efficiency by introducing BFF and CFF. These techniques enable fine-grained fusion of similar KV cache blocks across requests or chunks, achieving up to ×4.38 compression without compromising accuracy. By significantly reducing KV cache transfers, our method reduces significantly the average number of blocks, and thus, allows to scale the serving capacity effectively under heterogeneous workloads. Theoretical analysis based on Poisson point processes provides insight into the rate-distortion trade-offs, and extensive empirical evaluations across multiple benchmarks and model sizes validate the practical benefits. Looking forward, we plan to explore the impact on serving throughput, using adaptive threshold tuning, integration with quantization, and pruning methods.

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
