# OpenReview forum: "Joint Encoding of KV-Cache Blocks for Scalable LLM Serving"
_TMLR — Rejected by TMLR_

### Review · Reviewer_FJru · 2026-02-21

**Summary Of Contributions:**

The paper proposes Fast-Fusion, a scheme for compressing KV-caches by merging similar blocks across requests (Batch Fast-Fusion / BFF) and input chunks (Chunks Fast-Fusion / CFF) using cosine similarity thresholds, extending prefix sharing to approximate matches. A recursive tree-based strategy identifies fusion candidates in O(B log B). The authors model the threshold exceedance rate via Poisson point processes and bound the resulting attention distribution drift. Experiments on Llama3.1-8B and Qwen2.5-72B report up to 4.38× KV-cache compression and ~40% throughput gains on a single-machine vLLM benchmark.

**Additional Comments:**

The core idea here is sound and the systems integration with vLLM is practical work. The privacy observation in Remark 4.3 and the rate-distortion framing are nice touches. The paper reads well overall, though Section 6.4's title has a typo ("Perfromance") and the conclusion largely restates the abstract.

Some bibliography entries need attention: MMLU and GSM8k should cite the original papers (Hendrycks et al., Cobbe et al.), not just HuggingFace dataset URLs. The blog post citation (Huijong Jeong & Kim, 2024) has inconsistent author name formatting.

| Dimension | Score |
|---|---|
| Soundness | 2 / 4 |
| Presentation | 3 / 4 |
| Contribution | 2 / 4 |

**Overall:** The idea is practical and the direction is right, but the evaluation has a structural hole at its center — accuracy and throughput were never jointly validated. Combined with no baselines, no error bars, and an abstract that oversells the results, this requires a little more effort for publication. A revision that closes the eval gap and adds honest comparisons would be much appreciated.

**Audience:**

Yes

**Audience Explanation:**

The motivating observation — that >94% of KV blocks exceed 0.8 cosine similarity, yet exact prefix sharing captures only a small fraction of this redundancy — is practically important and well-demonstrated (Figure 1a). The jump from exact prefix matching to approximate block fusion is natural and addresses a real limitation in current serving systems like vLLM. Having both prefill (CFF) and decode (BFF) variants, integrated with vLLM's block table abstraction, makes this relevant to the systems/ML intersection that TMLR serves. The rate-distortion insight that compression scales with batch size at fixed accuracy threshold is also a useful observation for practitioners.

**Broader Impact Concerns:**

None beyond what's standard for inference optimization work. Remark 4.3's observation about reduced timing side-channel leakage relative to prefix sharing is a positive note, though not rigorously evaluated.

**Claims And Evidence:**

No

**Claims Explanation:**

The central issue is that accuracy and throughput have never been measured together. The paper is transparent about this — Sections 6.1–6.3 fuse and replicate blocks *without freeing memory*, while Section 6.4 frees blocks *without rescaling*. A joint evaluation "requires a dedicated kernel, which we keep in mind for future work." For a paper whose value proposition is simultaneous memory savings and accuracy preservation, this is a significant gap.

Beyond this structural problem:

- The abstract claims the method outperforms "recent structured and adaptive compression baselines," but no such comparison appears anywhere. Every experiment compares only against uncompressed vLLM.

- "4.38× with negligible accuracy loss" cherry-picks the best case (Qwen2.5-72B, batch 64, one dataset). Table 1 shows GSM8K F1 collapsing from 62.46 to 41.47 at ~3× compression on Llama3.1-8B. The paper needs to present the tradeoff curve honestly rather than advertising one corner of it.

- No result in the paper includes variance or is reported across multiple runs. Many accuracy differences under discussion are <1 F1 point — meaningless without error bars.

- The end-to-end evaluation uses 100 requests on random data, one model, one machine. Figure 7(d) shows throughput dipping *below* baseline for short outputs, which goes unmentioned. The ~40% headline is a single data point at maximum output length.

- The theoretical analysis (Propositions 5.1 and 5.3) rests on independence assumptions that clearly don't hold for correlated KV blocks, and the attention drift bound is never numerically evaluated at operating thresholds. It's unclear whether it gives useful values or is vacuous.

**Requested Changes:**

**Must-fix:**

1. Joint end-to-end evaluation. Fuse, free, and rescale in the same pipeline. Report both accuracy and throughput from the same runs. Without this, the paper's two claims (memory savings + accuracy preservation) are each validated under conditions where the other isn't operative.

2. Add comparisons to at least one or two competing KV-cache compression methods (e.g., KVQuant, MiniCache, or an eviction baseline). Remove the "outperforming recent baselines" claim from the abstract until this exists.

3. Report standard deviations or confidence intervals for all accuracy numbers across multiple runs. This is especially important for Table 1 and Table 2 where many differences from baseline are within plausible noise ranges.

4. Reframe the headline compression numbers. Present the full tradeoff curve prominently — don't lead with the best-case corner. The current framing in the abstract is misleading.

**Should-fix:**

5. Ablation on keys-only vs. values-only fusion. The paper fuses both but never justifies why both are necessary or whether one dominates.

6. Acknowledge and discuss the throughput regression for short output lengths visible in Figure 7(d). When does BFF help vs. hurt?

7. Evaluate Proposition 5.3's bound numerically at your operating thresholds. If the bound is loose, say so. If it's tight, that strengthens the theory section considerably.

8. Add perplexity on a standard benchmark (WikiText or similar) as a general quality metric alongside task-specific F1.

---

### Review · Reviewer_MGgW · 2026-02-23

**Summary Of Contributions:**

This paper introduced a notion termed the joint encoding of KV-cache blocks, which is a form of compression that "fuses" similar KV cache blocks into shared representations that are reused within (CFF) and across requests (BFF).

Strengths
- The high-level algorithmic design choices align with common serving engines like vLLM
- Evaluation covers multiple models and datasets

Weaknesses
- The practicality in real-world serving scenarios (where requests are more heterogeneous, of which the KVs might exhibit less similarity) was not studied
- The evaluations on end-to-end serving performance seem more like a simulated study of the performance upper-bound rather than an actual implementation
- Missing comparisons with related work

**Audience:**

Yes

**Audience Explanation:**

The memory bottleneck imposed by the KV cache is one of the most significant issues that serving engines face. Unlike existing literature for KV compression, this paper explores a different avenue through fusing the KV cache based on latent similarity. This approach, while not entirely novel, opens a new design space for memory optimizations. Even though I have issues with the current presentation and evaluation methodology of this paper, the theoretical modeling of KV cache similarity is indeed interesting and could inspire future research.

**Claims And Evidence:**

No

**Claims Explanation:**

- The motivation makes sense but seems a little arbitrary. Need to study on more dataset and model combinations. If an inference engine is serving requests from different distributions (e.g., some requests on math reasoning, some on general QA, some on coding), would we still see the similarity?
- I liked the premise of the paper, but the delivery of the evaluation feels underwhelming. For example:
    + Even though the authors performed a complexity analysis of the fusion strategy, the evaluation didn't seem to cover the latency overhead for operations like doing block similarity computations. Does computing cosine similarity itself involve memory-bandwidth bound operations? Does it happen on the CPU or GPU? Whether this work scales is, as a result, unclear.
    + It seems like there's a significant disconnect between the evaluations of end-to-end serving and the other evaluations. Indeed, as the authors noted in the beginning of the results section, specialized kernels are required to evaluate the efficacy of BFF and CFF in a real serving scenario. Without them, the effectiveness of BFF and CFF remains speculative. The paper mentions "Sections 6.1-6.3 assess compression and accuracy, by fuse and replicate the rescaled cache across all fused blocks (without freeing blocks). Section 6.4 examines the end-to-end latency by fusing and freeing blocks (without rescaling)". Why is there a gap between the two experiment methodologies?
- There is no quantitative comparisons with other "compression" baselines that reduces the memory footprint of the KV cache. The paper claims that "most prior methods rely on... limiting scalability and flexible integration into modern LLM serving pipelines". However, the main results (e.g., Fig. 4) suggest that non-negligible accuracy start to occur at a relatively modest compression ratio, which makes it difficult to appreciate the practicality of joint-encoding and the motivation for why it is superior to existing work (e.g., quantization, low-rank, or systems-side approximate KVs reusing). Further, BFF/CFF also relies on a pre-set threshold. Even though the authors noted (in remark 4.2) that this can be self-adjusted on-the-fly based on systems signals, no actual implementation is provided, which weakens the claim. For optimizations that are not accuracy-preserving, it's quite important to perform more thorough microbenchmarks on the efficiency-efficacy tradeoffs.
- The original KVs are discarded to save memory. However, if the current compression is too aggressive, and a serving system would want to raise the threshold, for the blocks that have already been fused and evicted, the information would be lost, and their KVs need to be recomputed from scratch. Can the authors clarify on this limitation?
- The latency numbers, TTFT in particular, in Fig. 7 seems unusually high. The dataset doesn't seem to be a long-context-focused workload, so it's unusual that the mean TTFT is in the scale of hundreds of seconds. More clarifications on the experiment details (e.g., hardware, engine arguments) are needed.

Presentation nits and minor questions
- The introduction section reads very redundant and is written more like a section in a survey paper. KV cache's memory bottleneck is a well-known issue -- more text should be spent on describing the joint-encoding scheme.
- Weird upper-casing in "nVidia OpenMathInstruct-2"
- What is the "random-ratio" in section 6.4?

**Requested Changes:**

- [Critical] Include more grounded experiments on the latency and compute overhead of the tree fusion process as a function of factors such as sequence length, batch size, block size, etc.
- [Critical] Add confidence intervals to figures
- [Critical] Include quantitative comparisons of BFF/CFF and other compression schemes (e.g., eviction approaches like StreamingLLM and H2O)
- [Critical] In Fig. 7, TTFT is evaluated against the output sequence length. However, TTFT is usually a function of the input sequence length and the batch size. I acknowledge that the authors were probably trying to demonstrate that BFF is more effective if there are more diverse blocks, but the presentation is confusing. Could the authors explain why TTFT correlates with output length here?
- Report accuracy and throughput in a more unified serving scheme rather than two setups

---

### Review · Reviewer_ctM4 · 2026-03-09

**Summary Of Contributions:**

This paper focuses on the KV cache compression and proposes joint encoding of KV-cache blocks, which fuses similar blocks across requests and input chunks into shared representations while preserving the standard cache structure.

**Audience:**

Yes

**Audience Explanation:**

KV cache compression is an important research area in the ML community.

**Claims And Evidence:**

Yes

**Claims Explanation:**

Experiments on Llama3.2 and Qwen2.5 demonstrates the effectiveness of proposed method.

**Requested Changes:**

1. Lack of Direct End-to-End Comparisons with SOTA Baselines

While the paper effectively highlights the limitations of existing rigid heuristics and exact-prefix matching frameworks, the experimental section predominantly compares the proposed Joint Encoding against an uncompressed baseline. To convincingly demonstrate the practical superiority of your approach, it is critical to include direct, end-to-end evaluations against recent SOTA KV-cache compression or sharing mechanisms discussed in your related work. Comparing against advanced quantization or dynamic sparsity methods would provide a much stronger validation of the proposed method's position in the current literature.

2. Absence of Critical Hardware and Implementation Details

The experimental setup lacks specific hardware configurations, merely stating the use of a "commodity GPU". Hardware specifics are paramount in memory-bound LLM serving optimizations. This paper must explicitly detail the exact GPU architecture used and its memory bandwidth, as these factors drastically influence the real-world performance of memory retrieval and matrix-matrix multiplications during the decoding phase. Furthermore, essential reproducibility details regarding the CUDA version and the specific vLLM framework version are missing and should be included.

3. Disconnect and Lack of Validation in Poisson Process Modeling

The theoretical analysis relies heavily on the assumption that the threshold exceedance rate follows a Poisson point process, which is derived from a Gaussian kernel approximation of similarity density. However, this mathematical modeling appears somewhat disconnected from the highly dynamic and heterogeneous real-world serving scenarios the paper targets. While Figure 3 illustrates this for specific layers (4, 11, 19, 25) of a single model, it is insufficient to prove that this Gaussian similarity assumption holds universally across all attention heads and at the deepest layers of varying architectures. This paper requires broader empirical validation to confirm that these theoretical rate-distortion bounds hold under the non-ideal, non-Gaussian distributions typical of diverse, real-world workloads.

---

### Decision · Action_Editor_XahX · 2026-04-27

**Recommendation:** Reject

**Audience:**

Yes

**Audience Explanation:**

The paper addresses a real and practically important problem - KV-cache memory pressure limits LLM serving concurrency, and exact prefix sharing captures only a fraction of available redundancy. The idea of fusing similar blocks by cosine similarity, integrated with vLLM's block table abstraction, is natural and well-motivated. The dual-mode design (BFF for decode, CFF for prefill) adds breadth, and the rate-distortion framing yields a useful insight: at fixed threshold, compression scales with batch size without additional accuracy cost.

**Claims And Evidence:**

No

**Claims Explanation:**

As detailed in the reviews, the paper is currently lacking some evidence for supporting the claims, such as evaluating accuracy and throughput jointly, as well as a more comprehensive evaluation against relevant baselines and making sure that the compression claims are holding in general and not for specifically chosen examples.

Addressing the comments brought up by the reviewers would significantly improve the paper. Given the relevance of the paper and the potential of the work, it will be great to see these in a future version.

**Resubmission Of Major Revision:**

The authors may consider submitting a major revision at a later time.